# Fibrosis and Conduction Abnormalities as Basis for Overlap of Brugada Syndrome and Early Repolarization Syndrome

**DOI:** 10.3390/ijms22041570

**Published:** 2021-02-04

**Authors:** Bastiaan J. Boukens, Mark Potse, Ruben Coronel

**Affiliations:** 1Department of Experimental Cardiology, Amsterdam University Medical Center, Amsterdam Cardiovascular Sciences, University of Amsterdam, 1105 AZ Amsterdam, The Netherlands; rubencoronel@gmail.com; 2Department of Medical Biology, Amsterdam University Medical Center, Amsterdam Cardiovascular Sciences, University of Amsterdam, 1105 AZ Amsterdam, The Netherlands; 3IHU Liryc, Electrophysiology and Heart Modeling Institute, Fondation Bordeaux Université, 33600 Bordeaux, France; mark@potse.nl; 4UMR5251, Institut de Mathématiques de Bordeaux, Université de Bordeaux, 33400 Talence, France; 5Carmen Team, INRIA Bordeaux—Sud-Ouest, 33400 Talence, France

**Keywords:** arrhythmias, Brugada syndrome, early repolarization pattern, fibrosis, conduction, repolarization

## Abstract

Brugada syndrome and early repolarization syndrome are both classified as J-wave syndromes, with a similar mechanism of arrhythmogenesis and with the same basis for genesis of the characteristic electrocardiographic features. The Brugada syndrome is now considered a conduction disorder based on subtle structural abnormalities in the right ventricular outflow tract. Recent evidence suggests structural substrate in patients with the early repolarization syndrome as well. We propose a unifying mechanism based on these structural abnormalities explaining both arrhythmogenesis and the electrocardiographic changes. In addition, we speculate that, with increasing technical advances in imaging techniques and their spatial resolution, these syndromes will be reclassified as structural heart diseases or cardiomyopathies.

## 1. Introduction

In Brugada syndrome and early repolarization syndrome patients, arrhythmias occur in the absence of gross structural abnormalities [1]. Both syndromes are characterized by abnormal deflections at the end or briefly after the QRS complex, which are referred to as J-waves. Based on these ECG characteristics, Brugada syndrome and early repolarization syndrome are referred to as J-wave syndromes and thought to share a pathophysiological mechanism [1,2,3].

The initially proposed mechanism underlying the Brugada syndrome involved the existence of regional differences in early repolarization [4]. These differences were initially described in the transmural plane of small blocks of canine ventricular myocardium, in which the subepicardial action potentials repolarize first [5]. The latter could set the stage for “phase-2 reentry” in the presence of a reduced net inward current, a loss-of-function mutation in the sodium or calcium channels, or by a gain-of-function mutation of the channel carrying the transient outward current [6]. This mechanism has been challenged by multiple experimental and clinical studies. These studies have shown that activation in the right ventricular outflow tract (RVOT) is discontinuous and delayed. This predisposes to excitation failure, ST-segment elevation, and the onset of reentry [7,8,9,10,11,12]. These findings have led to the now-accepted view that abnormal conduction in the RVOT underlies the Brugada syndrome [13,14]. This classifies the syndrome as a cardiomyopathy rather than a channelopathy [15]. For the early repolarization syndrome, the current view is still—as indicated by the name—that the syndrome is caused by enhanced local early repolarization [1]. However, we have recently shown that in a young patient local conduction delay caused by structural abnormalities led to the early repolarization syndrome [16]. Removal of the region containing the abnormalities cured the syndrome, indicating that abnormal conduction may be the mechanism causing early repolarization syndrome similar to the Brugada syndrome.

In this review, we will discuss how abnormal conduction explains the overlap in arrhythmogenic mechanisms in Brugada syndrome and early repolarization syndrome and how differences between the syndromes are explained. We speculate that the J-wave syndromes represent a part of a larger spectrum of cardiomyopathies.

## 2. Abnormal Conduction in RVOT in Brugada Syndrome

Structural abnormalities caused by interstitial fibrosis play an essential role in the generation of ST segment elevation and the onset of arrhythmias in Brugada syndrome patients [10,17,18,19]. These structural changes lead to a localized branching and converging myocardial network, very much resembling that of the network of surviving fibers in infarcted myocardium [8,20]. This may cause activation delay at the branching point (Figure 1A) and even excitation failure (Figure 1B) when the depolarizing current generated by the proximal myocardium is too small to activate distal myocardium (current-to-load mismatch) [21,22]. The ratio of the current generated and the minimum current required to maintain propagation is referred to as the safety factor for cardiac conduction and is also referred to as conduction reserve [22,23,24]. Sodium channel blockade decreases the safety for conduction and promotes conduction block at branching sites. This effect of sodium channel blockade forms the basis of the provocation test and explains the increased prevalence of mutations in the gene (*SCN5A*) encoding the cardiac sodium channel in the population with Brugada syndrome [25]. Conduction block at branching points delays local activation, causes fractionation in local unipolar electrograms [26], and may set the stage for reentrant arrhythmias (Figure 1C). Indeed, fractionated local electrograms have been recorded from patients with Brugada syndrome, although the duration of these fractionations does not cover the entire phase of ST-elevation [9,10,11]. The explanation for this apparent discrepancy is that the tissue beyond the tissue branching points is not activated at all (excitation failure) while being excitable and coupled to the proximal tissue. As a consequence, a potential difference occurs between the unexcited and adjacent activated tissue and generates a systolic current that is visible in the ECG as ST-segment elevation [12]. Fractionated potentials as a reflection of late activation thus occur merely at the end of the QRS complex and do not necessarily have to extend throughout the duration of the ST elevation.

In Brugada syndrome, arrhythmias are initiated predominantly in the RVOT [27]. In 2013, we have proposed that the intrinsic conduction reserve in the RVOT is lower than in the remainder of the heart because of its developmental origin, which would facilitate conduction delay and block in the presence of structural abnormalities [28,29]. The RVOT is often the source of arrhythmias, and the initiating premature activation can be the result of the arrhythmogenic substrate as described above [21].

## 3. Overlap in Mechanism of Brugada Syndrome and Early Repolarization Syndrome

### 3.1. Structural Abnormalities and Fractionated Potentials

Structural abnormalities have been reported repeatedly in Brugada syndrome patients [8,10,17,19]. Fibrosis is found in both the subendocardium [30] and subepicardium [10]. Whether fibrosis in Brugada syndrome patients is more prominent in the RVOT than in the remainder of the heart has not yet been thoroughly investigated. However, it has become clear that hearts from Brugada syndrome patients are histologically different from controls [31]. This suggests that the Brugada syndrome is a structural disease rather than a channelopathy [15]. In the ECG of patients with a structural heart disease, e.g., arrhythmogenic ventricular cardiomyopathy, Chagas disease, or Uhl’s disease, slurring or notching at the end of the QRS complex is often present and is referred to as the early repolarization pattern [32,33]. The early repolarization pattern is also present in the ECG of some Brugada syndrome patients [34]. Patients with the early repolarization pattern and with ventricular arrhythmias who, after standard clinical evaluation, appear to have a structurally normal heart are diagnosed with early repolarization syndrome [1]. However, epicardial mapping of these patients often reveals sites with fractionated electrograms [34,35], which are indicative of a structural substrate [26]. Ablation of these sites prevents ventricular fibrillation, indicating that these regions are required for arrhythmogenesis [9].

We recently reported the first case of early repolarization syndrome in which localized fibrosis in the right ventricular inferior wall caused ventricular fibrillation and the early repolarization pattern in the ECG [16]. Echocardiography and magnetic resonance imaging (MRI) did not reveal gross structural abnormalities in this patient. This leaves room for the possibility that other patients with the early repolarization syndrome have structural abnormalities as well, which have remained undetected so far. Currently, echocardiography, computed tomography, and MRI are used to assess structural and functional abnormalities in patients suspected of structural heart disease [15]. These techniques are presently not sensitive enough to detect transmural fibrosis in Brugada syndrome patients (and also not in the patient with early repolarization syndrome) [8,16]. A biopsy of the myocardium would allow detection of transmural fibrosis by conventional histology using a Picro-Sirius-red or Masson’s trichrome staining. Electroanatomic voltage mapping could designate the location for tissue sampling by identifying regions with low-amplitude fractionated potentials [36]. Establishing the causal relation between transmural fibrosis and fractionated potentials in patients with the early repolarization syndrome will foster radiofrequency ablation as a future antiarrhythmic therapy in patients with the early repolarization syndrome [37].

### 3.2. J-Waves in Local Electrograms and Excitation Failure

Late activation appears to be associated with pronounced local J-waves because the notch in the local epicardial action potential in late activated tissue is exposed in the ST segment [16,37,38]. This type of electrogram shows large R-waves—indicating late activated myocardium—followed by a positive J-wave as seen in Figure 2A (upper panel). These morphological features are often observed in local electrograms in early repolarization syndrome patients [16,38]. Similar J-waves that follow an R-wave (in a unipolar electrogram) can be found in Brugada syndrome patients as well, although they are often described following an S-wave [11,34]. Such J-waves may appear in tissue undergoing excitation failure caused by current-to-load mismatch whereby fractionated potentials precede a monophasic potential, imposing as ST elevation in the extracellular electrograms [21,39].

Excitation failure in early repolarization syndrome may manifest also in a different manner. Figure 2A shows a simulated electrogram from late-activated subepicardium of the right ventricular free wall before and after excitation failure (gray squares represent tissue with excitation failure). A notching of the QRS complex was seen, although the J-wave disappeared when excitation failure occurred. The same configuration was recorded from the right ventricular wall of a heart of a patient with early repolarization syndrome (Figure 2B). We speculate that in the patients at this location, excitation failure occurred leading to this electrogram morphology.

### 3.3. Generation of the J-Wave in the Electrocardiogram through Structural Abnormalities

If the compromised tissue—distal to the isthmus where excitation failure occurred—is located at the end of the normal activation pathway, it will not be activated at all, even though this tissue is excitable. The RVOT is normally the last part of the heart to be activated. As a consequence, a large systolic intracellular electrotonic “injury” current will flow from the activated tissue towards the unexcited tissue (located in the RVOT) and will generate a monophasic extracellular potential [11,39]. This potential will also be represented on the right precordial leads (or leads placed more cranially) of the ECG in the form of ST-elevation typical for Brugada syndrome [12]. However, if the compromised tissue is located in myocardium that is activated relatively late in the QRS complex but not at the very end of the normal activation pathway (not in the RVOT, but in the LV free wall or RV free wall), the initially unexcited tissue will be activated with a delay through a secondary activation route. As a consequence, a short-lasting systolic intracellular electrotonic current will flow from the activated tissue towards the unexcited tissue that will generate a short-lived positive extracellular potential. This potential will be represented on the according leads of the ECG as a fractionated QRS complex or J-wave in the ECG. This constitutes the typical early repolarization pattern.

### 3.4. Paradoxical Effect of Sodium Channel Blockade

Sodium channel blockade augments ST elevation in the Brugada syndrome but tends to suppress QRS slurring in the early repolarization syndrome. This paradox could hint towards a separate mechanisms giving rise to the ECG abnormalities in the different syndromes. However, this difference is likely related to the availability of alternative activation routes to the myocardium distal to the conduction block created by sodium channel blockade. In Brugada syndrome, such an alternative route is largely absent in the RVOT and the distal myocardium is, therefore, not excited, although it is excitable [21]. This creates the characteristic prolonged ST-elevation generated by coupling excited tissue to unexcited tissue (similar to the genesis of a monophasic action potential) [43]. In the heart with an early repolarization pattern, the structural abnormality is positioned in the free wall of the left or right ventricle, where alternative activation routes to the blocked myocardium are present. The sodium channel blockade, in this case, transforms the delayed activation into activation block. In addition, because the myocardium is not the terminally activated tissue, ajmaline-induced conduction slowing in the rest of the heart may delay activation in the rest of the heart beyond that of the structurally altered tissue. Experimental evidence for this mechanisms has been supported by computer simulations (Figure 3) [44].

## 4. Temporal Variability in Signs and Symptoms in Both Syndromes

Patients with Brugada syndrome or with the early repolarization syndrome do not always show the characteristic signs pertaining to the respective syndromes. The typical ST-T changes of Brugada syndrome are not always present and may only be observed after provocation with an intravenous sodium channel blocker (Ajmaline) or after a full-stomach test [45]. Even without provocative maneuvers, the symptoms and signs may wax and wane [46]. Similarly, the electrocardiographic changes belonging to the early repolarization pattern are unstable [1] and are modulated by exercise and sodium channel blockade. Small alterations in conduction reserve over time could explain the wax and wane of symptoms.

The reduction in conduction reserve at the site of source-sink mismatch can be modulated by various ionic currents other than the sodium current (I_Na_) [47]. An increase in the density of the transient outward current (I_TO_) or a decrease in that of the calcium current (I_CaL_) will reduce conduction reserve and hamper conduction across the site of tissue expansion and will exacerbate the conduction abnormality and the consequent ST elevation [21]. The densities of these currents fluctuate under influence of, e.g., heart rate, autonomic nervous system, diet, and circadian rhythm.

Drugs and compounds other than class I drugs have sodium channel blocker properties. For example, dietary fish oil consumption leads to sodium channel inhibition [48]. A meal containing fatty fish, therefore, may reduce conduction reserve [49] and provoke the Brugada sign. We speculate that this contributes to the full stomach effect that has been used as a provocation test for the Brugada sign [45]. Certainly, the postprandial parasympathetic surge may also be of importance.

But even without the use of sodium-channel blockers, the sodium channel can have reduced function. The upstroke velocity of the cardiac action potential depends on the resting membrane potential and therefore on the coupling interval between the premature activation and the previous activation. If the premature activation encroaches on the repolarization phase of the previous activation, the take-off potential of the premature beat is less negative, and subsequently fewer ion channels are available. However, if this is related to an increased sympathetic drive the influence of the calcium channel kicks in with contrary effects [50].

Hoogendijk et al. have shown in a computer model of a heart with structural abnormalities that a decrease in I_CaL_ also is capable of provoking the Brugada sign [21]. This effect is based on the observation that in structurally altered myocardium, the potential difference between activated and non-activated tissue may be maintained by the plateau of the action potential [51], if phase 0 has passed, and may even lead to propagation across a region of impaired conduction. Conversely, a temporary decreased calcium current (use of calcium channel blocking drugs and decreased sympathetic tone) may provoke activation block, if structural abnormalities are present.

The simulations by Potse et al. [21] have also demonstrated the modulatory effect of the transient outward current. If I_TO_ is increased, conduction across a branching structure may be hampered [51], and thus may lead to the Brugada sign. Typical circumstances in which the I_TO_ is increased are a slow heart rate (during sleep, increased parasympathetic tone). Conversely, inhibition of I_TO_ may exert an antiarrhythmic effect or decrease the Brugada sign. Because I_TO_ is particularly highly expressed in the subepicardial myocardium [52], the effects of I_TO_-modulation are expected particularly in the subepicardium.

## 5. Conclusions and Future Perspectives

Brugada syndrome is primarily caused by subtle, clinically undetectable, structural abnormalities in the RV and RVOT [15] and therefore should be considered as a cardiomyopathy. We speculate that at least a subgroup of the patients with the Early Repolarization Syndrome also display similar subtle structural myocardial abnormalities in regions other than the RVOT [16]. The common electrophysiological mechanism is formed by conduction abnormalities, caused by current-to-load mismatch. The modulatory effects of sodium channel inhibition and arrhythmogenesis can be explained by the same mechanism [10]. Although we cannot exclude the possibility that some of the patients with early repolarization syndrome have repolarization abnormalities (independent of altered conduction) the definition of J-wave syndromes should not include the criterion of absence of structural abnormalities [1]. The inability to detect these localized structural changes in the clinic is the reason that patients with a J-wave syndrome are thought to have structurally normal hearts [53]. Figure 4 shows the relation between the detection method and the diagnosis of various syndromes. We expect that with the increasing extent of the structural abnormalities (by ageing, comorbidities, or possibly by previous cardiac surgery), the J-wave syndromes will eventually become incorporated in the spectrum of structural heart diseases [15]. Similarly, with the expected increasing resolution of the imaging techniques, we expect that the same patients that are now diagnosed with one of the J-wave syndromes will be diagnosed with structural heart disease in the near future. The latter is relevant for patient management as a structural origin of arrhythmias provides a rationale for radiofrequency ablation as antiarrhythmic treatment [37].

## Figures and Tables

**Figure 1 ijms-22-01570-f001:**
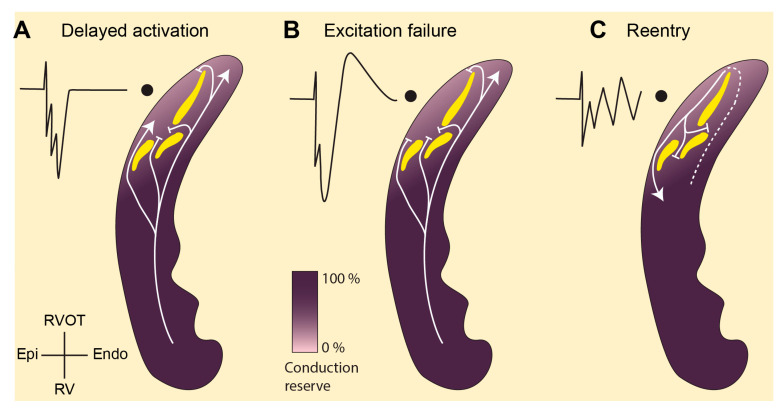
Abnormal conduction in the right ventricular outflow tract (RVOT) as explanation for the Brugada syndrome. (**A**). The structural abnormalities (yellow) in the RVOT separate activation fronts causing fractionated unipolar electrograms. (**B**). The abnormalities may also set the stage for current-to-load mismatch leaving part of the myocardium unexcited. This myocardium acts as a current sink during systole and receives current from adjacent—excited—tissue causing unipolar electrograms with ST segment elevation (J-point). (**C**) Current-to-load mismatch may also generate unidirectional block and the onset of reentry. The lighter shade of purple in the RVOT indicates a lower conduction reserve resulting from a distinct embryonic origin [28]. RVOT, right ventricular outflow tract; RV, right ventricle; Endo, endocardium; Epi, epicardium.

**Figure 2 ijms-22-01570-f002:**
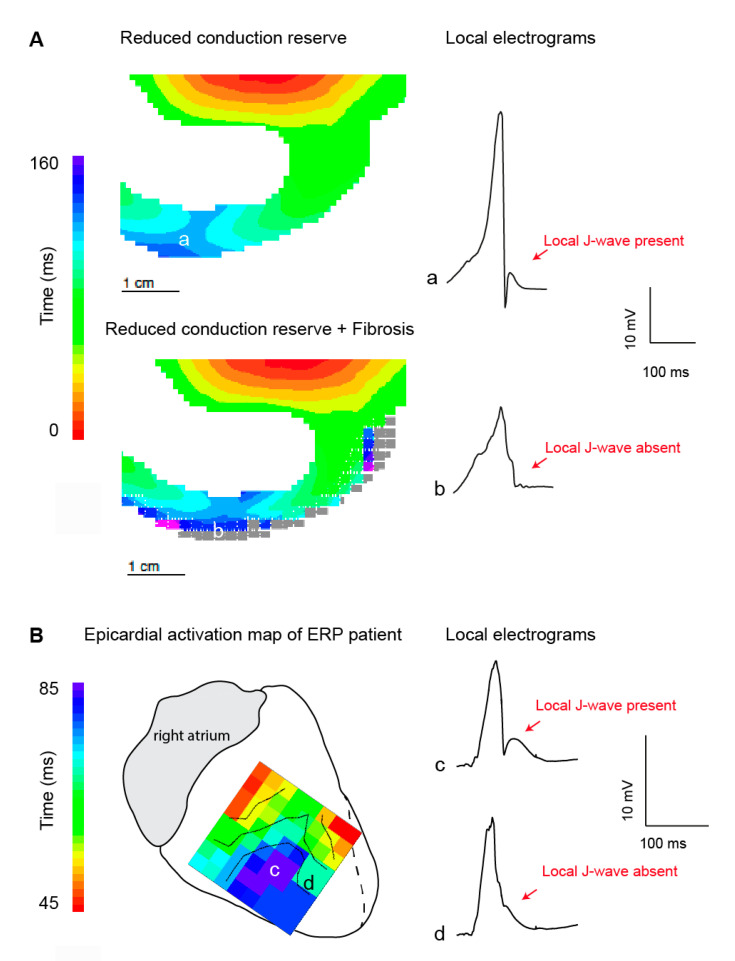
Local J-wave and excitation failure in relation to the early repolarization syndrome. (**A**). Simulated activation maps and extracellular potentials during sodium channel reduction (30%) in the absence (upper) and presence (lower) of structural abnormalities. Propagation and extracellular potential were simulated using a similar approach as published previously [12]. In brief, propagating action potentials (AP) were simulated with a monodomain reaction-diffusion equation, using software that has been described previously [40]. Ionic currents were computed with the Ten Tusscher-Noble-Noble-Panfilov model for the human ventricular myocyte [41], which distinguishes subendocardial, mid-myocardial, and subepicardial cell types. Computation of extracellular potentials (electrograms) from the simulated membrane potentials was based on the bidomain model for cardiac tissue [42]. Delayed activation in location a resulted in a pronounced J-wave in the corresponding local electrograms. In the presence of structural abnormalities excitation failure occurred and the local J-wave disappeared. Panel (**B**) shows the epicardial activation pattern at the right inferior wall measured during open chest mapping [16]. The delayed activated myocardium (in blue) showed unipolar electrograms with (c) and without (d) a local J-wave. We speculate that excitation failure occurred at location d. ERP, early repolarization pattern.

**Figure 3 ijms-22-01570-f003:**
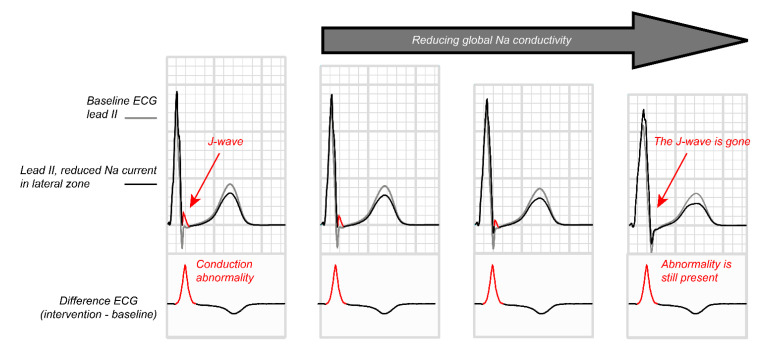
Inferolateral J waves and their attenuation by sodium channel blockers. The effects of reduced Na current, reduced coupling, and increased transient outward current on inferolateral J waves were evaluated with computer simulations using a detailed model of the human heart and torso [44]. Propagating action potentials were simulated with a monodomain reaction-diffusion model. At 1-ms intervals the simulated transmembrane currents were inserted in a torso model and a static bidomain problem was solved to obtain the electrocardiogram (ECG). Global reduction in Na-channel conductivity deformed and prolonged the QRS complex, masking the inferolateral J-waves. Subtraction of the ECGs with and without local intervention showed that the ECG change associated with the local reduction in Na current was not reduced, but was only hidden by the expanding QRS complex.

**Figure 4 ijms-22-01570-f004:**
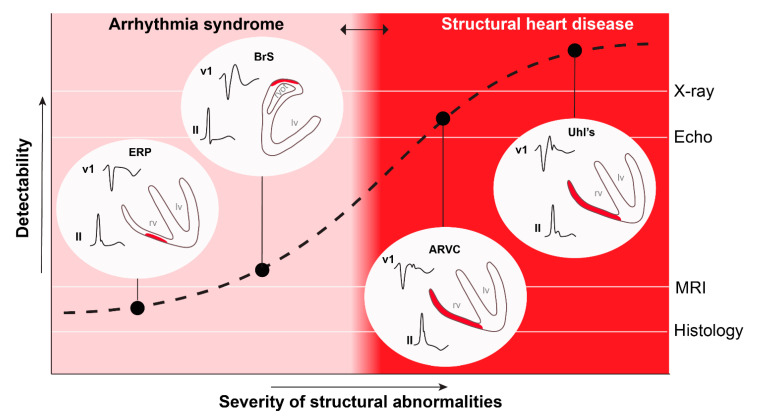
The definition of structural heart disease is based on the ability to detect structural abnormalities. An increased spatial resolution of the detection technique will lead to a leftward shift of the red zone. ERP, early repolarization pattern; BrS, Brugada syndrome; ARVC, arrhythmogenic right ventricular cardiomyopathy; Uhl’s, Uhl’s disease; MRI, magnetic resonance imaging; Echo, echocardiography. This figure was used with permission from [54].

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
