# Peer review of "Fibrosis and Conduction Abnormalities as Basis for Overlap of Brugada Syndrome and Early Repolarization Syndrome"

_ijms, 2021, doi:10.3390/ijms22041570_

Round 1
Reviewer 1 Report
The authors comprehensively describe the latest knowledge of fibrosis and conduction abnormalities in Brugada syndrome (BrS) and early repolarization syndrome (ERS). The reviews are carefully organized so it's easy to understand. But it requires minor revisions before the publication. The details are as follows.
- The title does not represent manuscripts contents. Please report more about the fibrosis in BrS and ERS.
- Also, please state about Uhl’s disease since Fig 3 included Uhl’s disease.
Author Response
Response 1. As asked by the reviewer we have more elaborately discuss the presence of fibrosis in Brugada syndrome and early repolarization syndrome patients in section 3.1 (page 4 lines 120-147, page 5 lines 148-150). We have replaced ‘fibrosis’ in the title with ‘ Structure’.
“Structural abnormalities have been reported repeatedly in Brugada syndrome patients [8,10,17,19] Fibrosis is found in both the subendocardium [30] and subepicardium [10]. Whether fibrosis in Brugada syndrome patients is more prominent in the RVOT than in the remainder of the heart is not thoroughly investigated. However, it has become clear that hearts from Brugada syndrome patients are histologically different from controls [31]. This suggests that the Brugada syndrome is a structural disease rather than a channelopathy [15]. In the ECG of patients with a structural heart disease - e.g. arrhythmogenic ventricular cardiomyopathy, Chagas disease or Uhl’s disease - slurring or notching at the end of the QRS complex is often present and is referred to as the early repolarization pattern [32,33]. The early repolarization pattern is also present in the ECG of some Brugada syndrome patients [34]. Patients with the early repolarization pattern and with ventricular arrhythmias with a structurally normal heart are diagnosed with early repolarization syndrome [35]. However, epicardial mapping of these patients often reveals fractionated electrograms [34,36], which are indicative of a structural substrate [26]. Ablation of sites showing local fractionated potentials prevents ventricular fibrillation indicating that these regions are required for arrhythmogenesis [37].
We recently reported the first case of early repolarization syndrome in which localized fibrosis in the right ventricular inferior wall caused ventricular fibrillation and the early repolarization pattern in the ECG [16]. Echocardiography and magnetic resonance imaging (MRI) did not reveal gross structural abnormalities in this patient. This leaves room for the possibility that other patients with the early repolarization syndrome have structural abnormalities as well, which have remained undetected so far. Currently, echocardiography, computed tomography and MRI are used to asses structural and functional abnormalities in patients suspected of structural heart disease [15]. These techniques are presently not sensitive enough for detection of transmural fibrosis in Brugada syndrome patients (and also not in the patient with early repolarization syndrome) [8,16]. A biopsy of the myocardium would allow detection of transmural fibrosis by conventional histology using a Picro-Sirius-red or Masson’s trichrome staining. Electroanatomic voltage mapping could designate the location for tissue sampling by identifying regions with fractionated potential with low amplitude [38]. Establishing the causal relation between transmural fibrosis and fractionated potentials in patients with the early repolarization syndrome will foster radiofrequency ablation as a future antiarrhythmic therapy in patients with the early repolarization syndrome [39].”
Response 2. We state about Uhl’s disease in section 3.1 on page 4 (lines 125-128):
“In the ECG of patients with a structural heart disease - e.g. arrhythmogenic ventricular cardiomyopathy, Chagas disease or Uhl’s disease - slurring or notching at the end of the QRS complex is often present and is referred to as the early repolarization pattern [32,33].”
Reviewer 2 Report
This is an interesting small scope review to discuss the potential mechanistic convergence of Brugada syndrome (BrS) and early repolarization syndrome (ERS). The topic and content will be very helpful for both clinical and basic science researchers to explore in this area. While the content is attractive, this review can be further enhanced with improved aesthetic presentation and a little more in-depth explanation. Suggestions are outlined below.
Major comments:
- The title suggests both “fibrosis” and “conduction abnormalities”. However, the review talk more explicitly on conduction abnormalities, while leaving fibrosis ambiguously implicated. There should be an assumption that unexcitable tissue is referring to fibrotic tissue. If fibrosis is suggested as the basis for both BrS and ERS, clinical detection and quantification on local fibrosis should be discussed.
- On the last sentence of the abstract, authors speculate that “these syndromes will be reclassified as structural and as cardiomyopathies”. Does it mean “structural cardiomyopathies”?
- It is strongly recommended to include a figure to explain how abnormal conduction in RVOT is contributing to BrS for section 2. The current section looks convoluted as it contains a lot of information.
- Section 3.3 is repeated.
- Authors should consider re-writing the first sentence of section 3.3. “The inexcited tissue will remain inexcited even though it is excitable” sounds contradictory in logics. It does not quite help even when authors spent some words to explain later. I also believe that it is should be “unexcited”, instead of “inexcited”.
Minor comments:
- In the abstract, line 16. IN should not be capitalized.
- In introduction, line 42. ERS was first used without indication.
- Same paragraph, line 44. Removal of the structurally abnormal myocardium sounds like the whole heart would be removed. I think authors mean regions of myocardium that is structurally abnormal?
- Section 32, line 109, “Black squares represent tissue with excitation failure”. In the figure, squares show grey in color.
Response 1.
Thank you for the positive assessment of our paper.
1. The title suggests both “fibrosis” and “conduction abnormalities”. However, the review talk more explicitly on conduction abnormalities, while leaving fibrosis ambiguously implicated. There should be an assumption that unexcitable tissue is referring to fibrotic tissue. If fibrosis is suggested as the basis for both BrS and ERS, clinical detection and quantification on local fibrosis should be discussed.
Response: We provide evidence that transmural fibrosis provides the substrate that is required for excitation failure to occur. We refer to the myocardium that “failed” to excite as unexcited tissue. The reviewer is incorrect in stating that the tissue is ‘unexcitable’. On the contrary, the tissue is excitable but is not excited, as a result of the branching tissue structure. The branching tissue structure can be the result of fibrosis, or of fat cell proliferation or any other non-myocardial cell. Thus, non-myocardial cells like fat or vessels are unexcitable and but can make part of an arrhythmogenic substrate as well. Transmural fibrosis in the RVOT appears to be a prerequisite for arrhythmias in patients with the Brugada syndrome. We have now included a schematic to illustrate these concepts. We have replaced ‘ fibrosis’ in the title with ‘structural abnormalities’.
We discuss clinical detection and quantification methods in the revised manuscript in section 3.1 on page 4 (lines 137-147) and page 5 (lines 148-150):
“Echocardiography and magnetic resonance imaging (MRI) did not reveal gross structural abnormalities in this patient. This leaves room for the possibility that other patients with the early repolarization syndrome have structural abnormalities as well, which have remained undetected so far. Currently, echocardiography, computed tomography and MRI are used to asses structural and functional abnormalities in patients suspected of structural heart disease [15]. These techniques are presently not sensitive enough for detection of transmural fibrosis in Brugada syndrome patients (and also not in the patient with early repolarization syndrome) [8,16]. A biopsy of the myocardium would allow detection of transmural fibrosis by conventional histology using a Picro-Sirius-red or Masson’s trichrome staining. Electroanatomic voltage mapping could designate the location for tissue sampling by identifying regions with fractionated potential with low amplitude [38]. Establishing the causal relation between transmural fibrosis and fractionated potentials in patients with the early repolarization syndrome will foster radiofrequency ablation as a future antiarrhythmic therapy in patients with the early repolarization syndrome [39].”
2. On the last sentence of the abstract, authors speculate that “these syndromes will be reclassified as structural and as cardiomyopathies”. Does it mean “structural cardiomyopathies”?
Response: Yes. We have adjusted the text as follows: “ ….. will be reclassified as structural heart diseases or cardiomyopathies.”
3. It is strongly recommended to include a figure to explain how abnormal conduction in RVOT is contributing to BrS for section 2. The current section looks convoluted as it contains a lot of information.
Response: As recommend by the reviewer we have generated a figure explaining how abnormal conduction in RVOT is causing the Brugada syndrome.
4. Section 3.3 is repeated.
Response: Thank you for noticing. We have removed the “repeated” section.
5. Authors should consider re-writing the first sentence of section 3.3. “The inexcited tissue will remain inexcited even though it is excitable” sounds contradictory in logics. It does not quite help even when authors spent some words to explain later.
Response: We have adjusted the sentence as follows (page 5 lines 171-173):
“If the compromised tissue – distal to the isthmus where excitation failure occurred – is located at the end of the normal activation pathway it will not be activated at all, even though this tissue is excitable. The RVOT is normally the last part of the heart to be activated.”
6. I also believe that it is should be “unexcited”, instead of “inexcited”.
Response: The reviewer is correct. It should be “unexcited” instead of “inexcited”. We have changed now us “unexcited” throughout the manuscript.
7. In the abstract, line 16. IN should not be capitalized.
Response: Adjusted.
8. In introduction, line 42. ERS was first used without indication.
Response: We have decided to not abbreviate ERS and ERP in the revised manuscript. Both terms are written fully throughout the text.
9. Same paragraph, line 44. Removal of the structurally abnormal myocardium sounds like the whole heart would be removed. I think authors mean regions of myocardium that is structurally abnormal?
Response: We have adjusted the sentence as follows (page 3, lines 77-78):
“Removal of the region containing the abnormalities cured the syndrome indicating that abnormal conduction may be the mechanism causing early repolarization syndrome similar to the Brugada syndrome.”
10. Section 32, line 109, “Black squares represent tissue with excitation failure”. In the figure, squares show grey in color.
Response: Thank you. We have changed “black” into “grey” in line 164.